# Cryo-EM structures of human STEAP4 reveal mechanism of iron(III) reduction

Wout Oosterheert[1], Laura S. van Bezouwen[1,2], Remco N. P. Rodenburg[1], Joke Granneman[1], Friedrich Förster[2], Andrea Mattevi [3] & Piet Gros [1]

Enzymes of the six-transmembrane epithelial antigen of the prostate (STEAP) family reduce $Fe^{3+}$ and $Cu^{2+}$ ions to facilitate metal-ion uptake by mammalian cells. STEAPs are highly upregulated in several types of cancer, making them potential therapeutic targets. However, the structural basis for STEAP-catalyzed electron transfer through an array of cofactors to metals at the membrane luminal side remains elusive. Here, we report cryo-electron microscopy structures of human STEAP4 in absence and presence of $Fe^{3+}$-NTA. Domain-swapped, trimeric STEAP4 orients NADPH bound to a cytosolic domain onto axially aligned flavin-adenine dinucleotide (FAD) and a single b-type heme that cross the transmembrane-domain to enable electron transfer. Substrate binding within a positively charged ring indicates that iron gets reduced while in complex with its chelator. These molecular principles of iron reduction provide a basis for exploring STEAPs as therapeutic targets.

[1] Crystal and Structural Chemistry, Bijvoet Center for Biomolecular Research, Department of Chemistry, Faculty of Science, Utrecht University, Padualaan 8, 3584 CH Utrecht, The Netherlands. [2] Cryo-Electron Microscopy, Bijvoet Center for Biomolecular Research, Department of Chemistry, Faculty of Science, Utrecht University, Padualaan 8, 3584 CH Utrecht, The Netherlands. [3] Department of Biology and Biotechnology 'L. Spallanzani', University of Pavia, 27100 Pavia, Italy. Correspondence and requests for materials should be addressed to P.G. (email: p.gros@uu.nl)

Assimilation of elemental iron is essential for the survival of all aerobic living species. Iron performs crucial functions in diverse physiological processes, ranging from DNA synthesis to oxygen transport to oxidative phosphorylation. Conversely, iron transport pathways are perturbed in cancer cells[1], which require high intracellular iron levels for rapid growth.

In mammals, iron mainly exists in its oxidized form ($Fe^{3+}$) bound to carrier protein transferrin in serum. However, iron uptake by cells requires ferrous iron ($Fe^{2+}$) because metal transporters specifically import divalent cations. First identified as antigens upregulated on the membranes of prostate cancer[2], enzymes of the six-transmembrane epithelial antigen of the prostate (STEAP) protein family catalyze iron(III) reduction[3–5] and thereby provide the crucial link between iron sequestering and iron import by mammalian cells. STEAPs are associated with metabolic diseases[6–8] and are over-expressed in several human cancers[2,9–11], underlining their physiological function in maintaining cellular iron homeostasis[6,12].

The STEAP family comprises four integral membrane proteins (STEAP1-4) that consist of a N-terminal cytoplasmic oxidoreductase domain (OxRD) and a six-helical transmembrane domain (TMD). They transfer electrons from intracellular NADPH to metal ions at the opposite side of the membrane through FAD and a single heme. Interestingly, STEAPs are the only described eukaryotic cytochromes that perform transmembrane electron transport using a single-heme ligand;[13] other reductases contain an axial diheme motif in their TMD, indicating that STEAP enzymes transfer electron across membranes through a unique cofactor arrangement. Previous studies suggest that the FAD cofactor might substitute and mimic the second, missing heme in STEAPs, because FAD interacts with the intracellular loops of the TMD[14,15], although the structural basis for this observation is lacking. Despite the association of STEAP proteins with severe diseases and although they are potential therapeutic targets[16–18], structural information is only available for the cytoplasmic NADPH-binding OxRD of human STEAP3[19] and rat STEAP4[14], but is absent for the cofactor and substrate-binding TMD of any STEAP.

To gain insights into the molecular mechanisms that govern the metalloreductase activity of STEAP proteins, we determine structures of human STEAP4 using single-particle cryo-EM. The trimeric structures reveal an aligned, inter-subunit NADPH-FAD-heme arrangement and suggest that chelated iron(III) binds in a cavity of basic residues to facilitate reduction.

## Results

**Biochemical characterization.** We screened the detergent stability of all human STEAP orthologs by fluorescence-size exclusion chromatography[20]. STEAP4 (also known as STAMP2[21] and TIARP[22]) retained heme throughout multi-step purifications (Supplementary Fig. 1a–c) and therefore emerged as a promising target for detailed characterization. For structural studies, we used a STEAP4 construct (deemed STEAP4$_{EM}$) that spans residues K18–Q456 (ΔN17ΔC3). STEAP4$_{EM}$ has similar cell-surface ferric-reductase activity as the full-length protein when expressed in HEK-cells (Supplementary Fig. 1d). We also measured the ferric-reductase activity of isolated, detergent-solubilized STEAP4$_{EM}$ and observed STEAP-specific formation of $Fe^{2+}$ in the presence of FAD and NADPH (Fig. 1a, Supplementary Fig. 1e, f). The calculated Michaelis-Menten constants ($K_M$) for $Fe^{3+}$-NTA, and NADPH were 13.7 μM, and 4.2 μM respectively, whereas the dissociation constant ($K_d$) for FAD was 1.2 μM. These values are comparable to those previously reported for STEAP3 in isolated cell membranes[15] and STEAP4 in cells[14], indicating that purified

STEAP4$_{EM}$ binds its substrate and cofactors with physiological affinities. Collectively, these experiments show that STEAP4 alone reduces iron and does not require any protein partners to function as a metalloreductase. We next determined the oligomeric state of detergent-purified STEAP4$_{EM}$ with multi-angle laser light scattering (MALLS), which revealed that the enzyme forms trimeric assemblies (Supplementary Fig. 1g). This finding is consistent with MALLS experiments on STEAP1[5], but inconsistent with the dimeric crystal structures of truncated OxRDs of human STEAP3 and rat STEAP4[14,19]. Thus, interactions formed by the TMDs control STEAP multimerization.

**Cryo-EM structure determination.** We optimized conditions for structure determination of STEAP4$_{EM}$ by employing a thermostability assay[23,24]. The addition of cofactors NADPH and FAD, combined with lowering the pH from 8.0 to 5.5, strongly stabilized STEAP4$_{EM}$ (Supplementary Fig. 1h) and led to an increase in melting temperature of 11 °C (from 45.3 to 56.3 °C). The higher stability at acidic pH may reflect a physiologically relevant condition, because STEAPs localize in endosomes[3,12,21,25]. We then determined structures of STEAP4$_{EM}$ in the presence of NADPH and FAD (cofactor-bound state) and in the presence of $NADP^+$, FAD, and $Fe^{3+}$-NTA (cofactor/substrate-bound state) using single-particle cryo-EM. The cofactor and cofactor/substrate-bound datasets yielded reconstructed density maps at 3.8 and 3.1 Å resolution, respectively (Fig. 1b, c, Supplementary Figs. 2–4, Table 1). We argue that the difference in resolution is mainly caused by microscope type (200 kV FEI Arctica vs. 300 kV FEI Krios) and not by sample quality. The density of the cytosolic OxRD was fitted to the crystal structure of rat STEAP4 (residues 20–195, pdb 2yjz), which exhibits 81% amino-acid residue identity to the human sequence. The model for the α-helical TMD, for which no template was available, was built de novo. The final models display good stereochemistry and are consistent with the EM density maps within the determined resolutions (Supplementary Figs. 5, 6, Table 1). Since both structures are virtually identical to 3.8-Å resolution (Fig. 1b, c, Supplementary Fig. 7) except for the substrate-binding site, we mainly focus on the higher 3.1-Å resolution cofactor/substrate-bound structure for the analysis.

**Overall architecture.** STEAP4 possesses a trimeric architecture (Fig. 1d–f). Each subunit contains six membrane helices (h1–h6), with helices h2, h3, h4 and h5 forming the cofactor-binding core of the protein that spans the (~32-Å) membrane bilayer (Fig. 1d). Residues between helices h3 and h4 form a α-helix (eh) that sticks 22 Å into the extracellular or endosomal milieu. The intracellular OxRD protrudes up to 46-Å distance into the cytoplasm and is structurally very similar to the crystal structure of the OxRD of rat STEAP4 (rmsd = 0.4 Å for 154 Cα atoms), even though the latter forms dimers instead of trimers (Supplementary Fig. 8). The trimeric arrangement observed in the cryo-EM structures is formed by interactions in STEAP protomers in both the OxRD and TMD layers (Fig. 1d–f), with a phospholipid molecule packed in between the TMDs of adjacent enzyme subunits, indicating that lipids stabilize the trimer (Supplementary Fig. 6c, d). Moreover, the OxRD of one STEAP4 chain is positioned beneath the TMD of an adjacent chain, yielding a domain-swapped arrangement, which suggests that STEAP1 (that lacks an intracellular OxRD) may become functional through heteromerization, as in STEAP1-STEAP2 complexes which co-express in cancers[17] and co-purify in detergent[5].

**Heme and FAD-binding sites in the TMD.** We observed density for one heme cofactor in each STEAP4 subunit (Fig. 2a). The

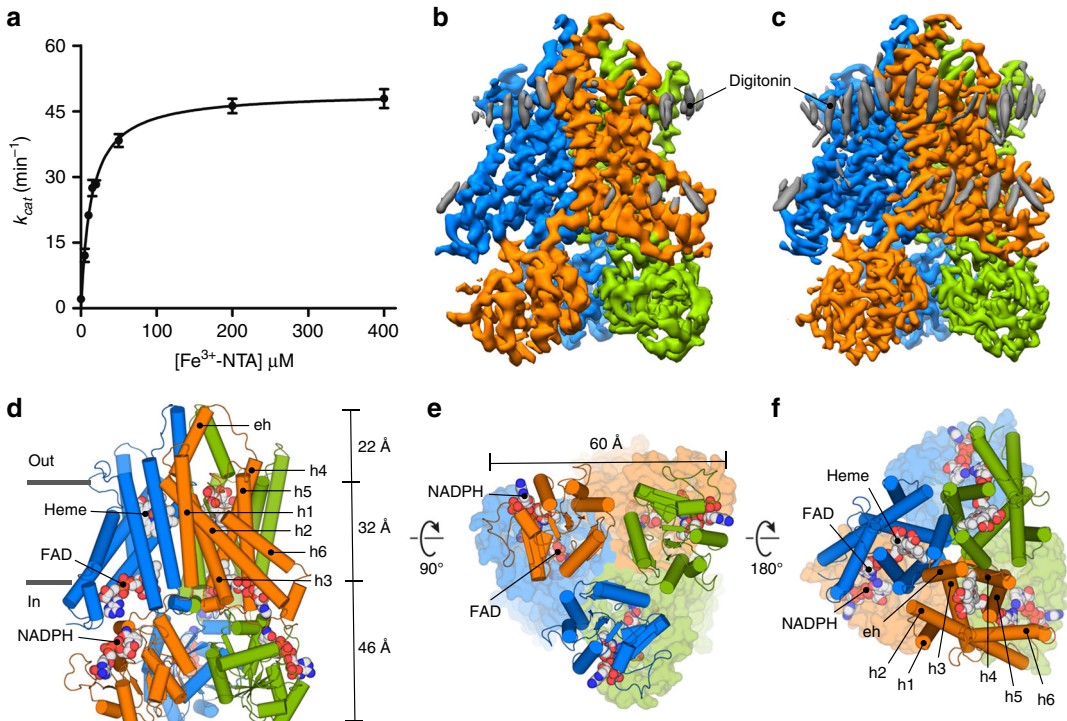

**Fig. 1** Catalytic activity and molecular architecture of human STEAP4. **a** Ferric-reductase activity of detergent-purified STEAP4$_{EM}$ with varying Fe$^{3+}$-NTA concentrations. STEAP4$_{EM}$ exhibits a $K_M$ of $13.7 \pm 0.8$ μM for Fe$^{3+}$-NTA and a $k_{cat}$ of $49.4 \pm 0.8$ min$^{-1}$. Error bars represent the standard deviation between triplicate measurements. **b** Sharpened density map of the cofactor-bound dataset at 3.8-Å resolution colored by chain. **c** Sharpened density map of the cofactor/substrate-bound dataset at 3.1-Å resolution colored by chain. The grey densities do not exhibit protein features but likely represent the sterol moiety of digitonin. **d**–**f** Trimeric arrangement of STEAP4, viewed parallel to the membrane as a sideview (**d**) and perpendicular to the membrane from the cytoplasm (**e**), and extracellular milieu (**f**). For the cytoplasmic and extracellular views, the domain in the background is shown as surface

heme is located in a protein pocket at the extracellular side of the TMD. The propionate arms of the porphyrin interact with R245, Y401 and R405, and point towards the extracellular space (Fig. 2b). The vinyl groups form hydrophobic interactions with several amino acids and orient towards the membrane center. Strictly conserved residues H304 and H397 of helices h3 and h5, respectively, coordinate the central heme iron, defining a hexa-coordinated porphyrin (Fig. 2a, b, e).

At the intracellular side of the TMD cavity formed by helices h2–h5, we found density for an FAD molecule (Fig. 2c), which is axially anchored in the membrane through its tricyclic flavin ring. FAD has an extended conformation and interacts with numerous conserved amino acids (Fig. 2d). Residue Q383 of helix h5 sandwiches the flavin ring onto K291 and G294 of helix h3, thereby making both faces of the ring inaccessible to other ligands. The flavin ring is further stabilized through interactions with D148, L255, F359, L362 and S366, whereas R290, S384, and K434 coordinate the phosphate moieties of FAD. The adenine base stacks with W140 of a domain-swapped OxRD, indicating that FAD bridges the intracellular and TM domains of adjacent STEAP4 subunits.

Does the orientation of FAD with respect to heme in the TM core allow for electron transport across the membrane? The dimethylbenzene moiety of the TMD-inserted flavin ring faces vinyl 4 (Fisher nomenclature) of the heme (Fig. 2e). The shortest distance between FAD and heme is 9.6 Å. Both molecules are within 4-Å distance of residue F359, which is located in the center of the TMD core of STEAP4 (Fig. 2e). Based on this analysis, we propose that electrons travel to the luminal side of the membrane through a FAD-F359-heme motif. STEAPs thus facilitate transmembrane electron transport by a single heme.

**Structural similarities with NADPH oxidases**. To compare the cofactor arrangement in the TMD of STEAP4 to that in other proteins with a ferric-reductase domain (FRD), we superimposed the STEAP4 structure to the crystal structure of the monomeric TMD of *Cylinodrospermum stagnale* NOX5[26], which functions as O$_2$ reductase and exhibits 17% sequence identity to STEAP4 in the region spanning helices h2–h5. STEAP4 and NOX5 share the same topology (rmsd 5.2 Å for 184 Cα atoms, Fig. 3a) and the heme of STEAP4 has a comparable orientation and coordination as the heme present in the extracellular membrane leaflet of NOX5 (Fig. 3b). Remarkably, the cytoplasmic leaflet heme of NOX5 binds at a similar height in the membrane as the flavin ring of FAD in STEAP4 (Fig. 3c). STEAPs evolutionary diverged from other FRD-family enzymes by losing the histidine residues that coordinate the heme in the cytoplasmic membrane leaflet[13]. Our structural analysis now supports the conclusion that the heme-binding site in the cytoplasmic membrane leaflet evolved into a flavin-binding site[14].

**Iron(III)-binding cavity**. Binding of substrate Fe$^{3+}$-NTA does not induce conformational rearrangements in STEAP4, because the densities of the cofactor/substrate and the cofactor-bound structures are virtually identical (Fig. 1b, c, Supplementary Fig. 7). Nevertheless, a prominent and highly significant density, positioned directly above the porphyrin in each protomer, is observed in a difference map between the two reconstructions (Fig. 4a). In the regular (non-difference) map, this density is weaker than we expected for Fe$^{3+}$ or Fe$^{3+}$-NTA (Supplementary Fig. 9), even though the cryo-EM sample was supplemented with Fe$^{3+}$-NTA at a concentration 50× higher than the $K_M$ (Fig. 1a). This suggests

**Table 1 Cryo-EM data collection, refinement and validation statistics**

| | STEAP4$_{EM}$—NADPH, FAD, heme bound (EMDB-0200) (PDB 6HD1) | STEAP4$_{EM}$—NADP$^+$, FAD, heme, Fe$^{3+}$-NTA bound (EMDB-0199) (PDB 6HCY) |
|---|---|---|
| **Data collection and processing** | | |
| Magnification | FEI Talos Arctica | FEI Titan Krios |
| Voltage (kV) | 200 | 300 |
| Electron exposure (e-/Å$^2$) | 45.5 | 47.4 |
| Defocus range (μm) | −0.8 to −3.0 | −0.8 to −2.0 |
| Pixel size (Å) | 1.029 | 0.813 |
| Symmetry imposed | C3 | C3 |
| Initial particle images (no.) | 421,438 | 1,098,075 |
| Final particle images (no.) | 209,075 | 255,144 |
| Map resolution (Å) | 3.80 | 3.09 |
| 0.143 FSC threshold | | |
| Map resolution range (Å) | 3.60–4.80 | 2.95–3.60 |
| **Refinement** | | |
| Initial model used (PDB code) | – | – |
| Model resolution (Å) | 3.84 | 3.12 |
| 0.5 FSC threshold | | |
| Map sharpening $B$ factor (Å$^2$) | −166 | −80 |
| Model composition | | |
| Non-hydrogen atoms | 11007 | 11007 |
| Protein residues | 1308 | 1308 |
| Ligands | 12 | 12 |
| $B$ factors (Å$^2$) | | |
| Protein | 110.1 | 96.3 |
| Ligand | 99.6 | 90.1 |
| R.m.s. deviations | | |
| Bond lengths (Å) | 0.009 | 0.010 |
| Bond angles (°) | 0.998 | 1.096 |
| Validation | | |
| MolProbity score | 1.54 | 1.47 |
| Clashscore | 3.40 | 2.66 |
| Poor rotamers (%) | 0.79 | 0.79 |
| Ramachandran plot | | |
| Favored (%) | 93.8 | 93.8 |
| Allowed (%) | 6.2 | 6.2 |
| Disallowed (%) | 0 | 0 |

that the substrate is coordinated heterogeneously or alternatively, that it is perturbed by electron radiation[27], which limits an accurate modeling of Fe$^{3+}$ or Fe$^{3+}$-NTA in the density. STEAP4 residues are positioned far beyond ~2.0-Å from the center of the difference density, indicating no direct interactions of Fe$^{3+}$ with residues of enzyme. Instead, we observed nine basic amino acids surrounding the difference density at 4.5–9 Å distances (Fig. 4b). These residues form a positively charged, ~16-Å wide ring (Supplementary Fig. 10) that is consistent with binding of Fe$^{3+}$, while it remains in complex with a negatively charged chelator like NTA or citrate. We propose that this ring-shaped arrangement of basic amino-acid residues provides a second coordination shell, which may facilitate metal reduction by controlling the position of chelated Fe$^{3+}$ with respect to the heme, and by modulating the polarity of the iron-chelator complex, thereby making Fe$^{3+}$ more susceptible to electron uptake. Thus, STEAP4 provides a favorable chemical environment for the reduction of its substrate.

**Electron transfer from NADPH to FAD**. NADPH binds in a similar orientation to the intracellular OxRD, as observed in the crystal structures of isolated OxRDs of human STEAP3 (pdb 2vq3), rat STEAP4 (pdb 2yjz) and an archaeal F$_{420}$H$_2$:NADP$^+$ oxidoreductase (FNO, pdb 1jay[28]) (Supplementary Figs. 8, 12). Although purified STEAP4$_{EM}$ displays catalytic activity, the nicotinamide ring of NADPH does not stack to a flavin ring in the observed structures; the closest flavin ring atom resides at

~18-Å distance in the adjacent STEAP4 TMD subunit (Fig. 5a). We performed enzymatic assays using flavin-mononucleotide (FMN) instead of FAD, which revealed that FMN may act as a cofactor in STEAP4-catalyzed iron reduction, albeit only at high, non-physiological concentrations (Supplementary Fig. 11). Taken together with the moderate $K_d$ of FAD, this suggests that FAD could be mobile. Superimposition of the STEAP4 OxRD onto FNO (rmsd = 0.9 Å for 120 Cα atoms), which has NADPH bound in a similar orientation with a stacked flavin-like F$_{420}$ molecule, suggests a potential NADPH-stacking site for a diffusible FAD in STEAP4 (Supplementary Fig. 12). In between both cofactor-binding sites, STEAP4 exhibits a cavity that is large enough to act as a putative tunnel for a passing FAD, without requiring large conformational protein rearrangements (Fig. 5a). To probe this, we mutated strictly conserved residue S138, which resides in the tunnel (Fig. 5b–d), to glutamine. The purified S138Q mutant retained heme and formed trimers (Supplementary Fig. 13), but displayed strongly impaired ferric-reductase activity. We postulate that the bulkiness of the glutamine side chain hinders the diffusion of FAD to NADPH, providing evidence for an electron transfer model in which the flavin ring diffuses between two sites in adjacent STEAP4 protomers to shuttle electrons from NADPH to heme (Fig. 5d).

## Discussion
Our biochemical and structural characterization of human STEAP4 elucidated that detergent-solubilized STEAP4 forms

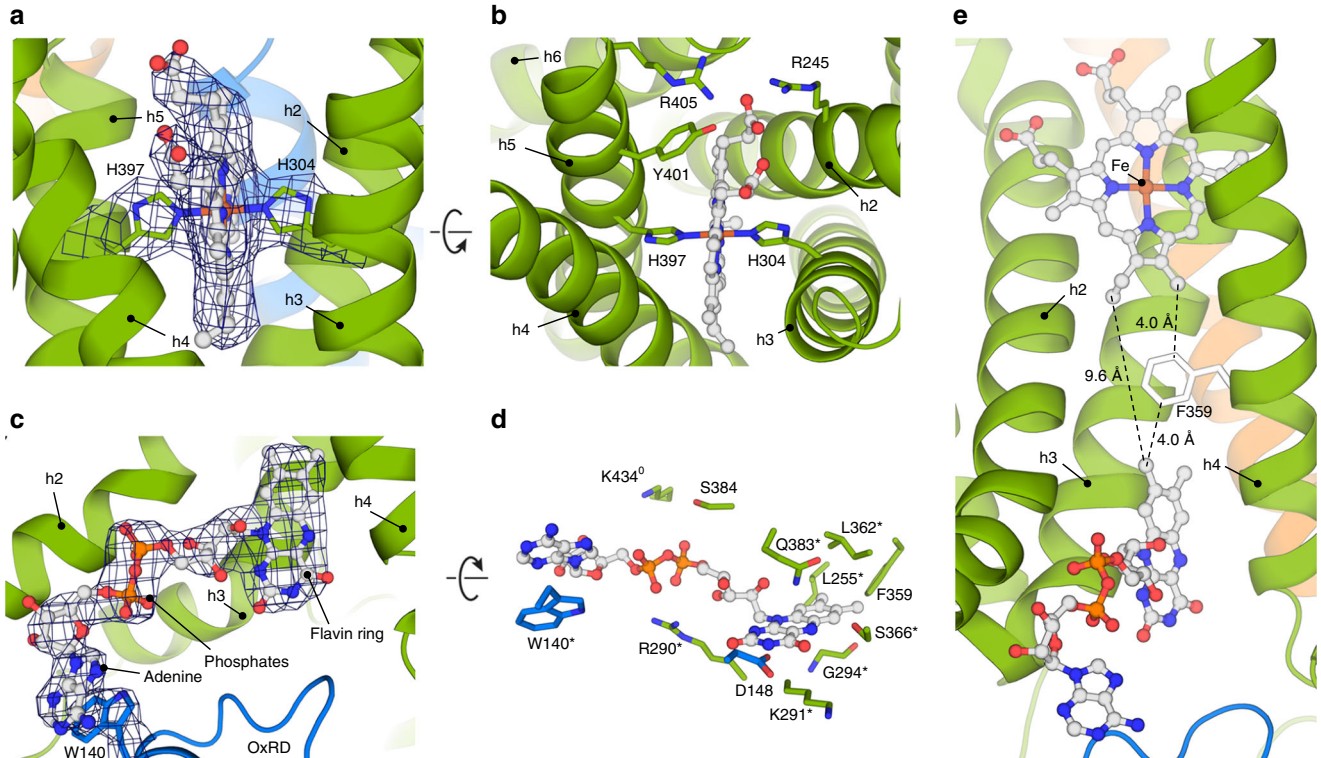

**Fig. 2** Cofactor arrangement in the membrane. **a** Heme coordination by residues H304 and H397. The density map is contoured at 8.2σ. **b** The heme-binding pocket in a STEAP4 subunit viewed from the extracellular milieu. **c** FAD-binding site at the intracellular side of the membrane. The FAD and W140 density are contoured at 8.2σ. Helix h5 resides in front of FAD and is omitted for clarity. **d** FAD-binding interface in STEAP4. Residues annotated with (*) display > 99% sequence conservation throughout the STEAP family, whereas residues annotated with (⁰) exhibit conservative substitutions throughout all STEAPs. **e** Orientation of FAD with respect to heme in the TMD of each STEAP4 subunit. Helix h5 is omitted from the figure

trimers that exhibit ferric-reductase activity in the presence of its redox cofactors, FAD and heme, and electron donor, NADPH, indicating that the enzyme does not require specific accessory proteins for catalysis.

Cryo-EM structures of STEAP4 in cofactor-bound and cofactor/substrate-bound states revealed the binding of FAD and heme molecules to the TMD of each protomer. The architecture of these cofactor-binding sites rationalizes previously reported biochemical and kinetic data. Residues H304 and H397, which coordinate the central heme iron (Fig. 2a, b), were reported by several studies to be imperative for heme binding and metal-ion reduction by STEAP proteins[9,15]. Strictly conserved residues R290, K291 and Q383 were identified through mutagenesis studies to participate in FAD binding;[15] the cryo-EM structures confirm direct interactions between FAD and these residues; in addition, the structures reveal several other FAD-interacting amino-acid residues (Fig. 2d). The minimum distance of 9.7 Å between FAD and heme indicates that both cofactors reside within electron-transfer distance[29]. Transfer of electrons across the membrane may follow a path through residue F359 positioned in between FAD and heme (Fig. 2e). STEAP orthologs harbor phenylalanines, tyrosines or leucines at the equivalent position, suggesting that the presence of an aromatic residue is not critical for transporting electrons across the membrane and that alternative routes through the protein may exist. Likewise, the residue located in between the diheme motif in the TMD of NOX5 is not strictly conserved as aromatic residue within the NOX family[26].

A difference map between EM-density reconstructions of STEAP4 with and without its substrate revealed that the iron substrate binds above the heme in a large cavity of nine basic amino-acid residues (Fig. 4b). Although the current data does not allow to accurately model the substrate in the density, it is obvious that binding of iron(III), by itself, in this electropositive environment is highly unfavorable. This implies that STEAPs reduce iron while it remains in complex with a negatively charged chelator. This structural implication is in agreement with findings from an earlier electrochemical study, in which the same conclusion was derived based on the measured redox potential of detergent-purified STEAP1[5]. Sequence comparison of 607 STEAP orthologs (see Fig. 4b legend) indicates that not all nine basic residues in the STEAP4 substrate-binding cavity in STEAP4 are highly conserved. However, all human STEAPs harbor at least six positively charged residues at the substrate-binding site, supporting a common mechanism for substrate binding and reduction. Although our current study focused on chelated iron(III) as substrate, we speculate that Cu(II)-chelator complexes are reduced via the same mechanism as iron(III).

In conclusion, the work presented here describes the structure-function analysis of a homotrimeric STEAP protein that contains both its OxRD and TMD. We propose that STEAPs reduce metal ions through the following steps: (1) NADPH binds to the cytosolic OxRD, (2) the flavin ring of FAD stacks on to the nicotinamide ring of NADPH and accepts a H⁻ ion, resulting in FADH₂. (3) FADH₂ diffuses to its TMD-anchored site in the adjacent protein subunit and releases electrons, one at a time, to heme, (4) heme then donates the electron to a chelated $Fe^{3+}$ or $Cu^{2+}$ ion. Subsequently, metal transporters import the reduced metal ion into the cell.

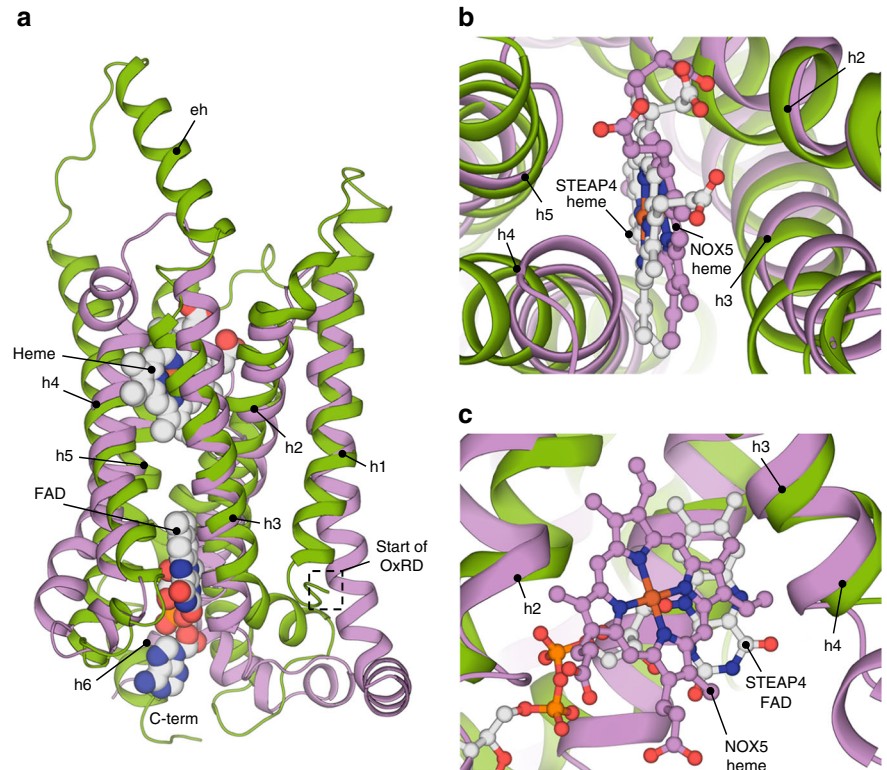

**Fig. 3** Superimposition of STEAP4 on the TMD of NOX5. **a** Global alignment of STEAP4 residues 195–454 (green) to the crystal structure of csNOX5[26] residues 207–412 (violet) shown parallel to the membrane in side view. Overlapping helices are annotated. **b** Architecture of the heme-binding pockets as viewed perpendicular to the membrane from the extracellular side of the TMD. **c** Arrangement of the flavin-binding cavity of STEAP4 that overlaps with the intracellular heme-binding pocket in csNOX5

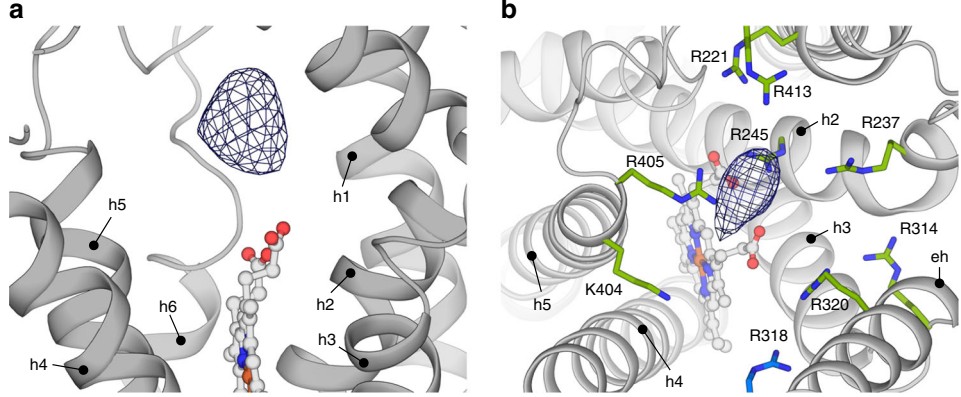

**Fig. 4** Substrate-binding site above the porphyrin. **a** Sideview of the extracellular side of the TMD of a STEAP4 protomer (shown in grey). The difference density is contoured at 18σ. **b** Positively charged amino acid ring that surrounds the substrate density, as viewed from the extracellular side of the membrane. Residue R318 belongs to an adjacent STEAP4 protomer. The difference density is contoured at 18σ. The shown residues display variable conservation as basic amino acids (percentage in brackets) throughout 607 STEAP orthologs:[15] R221 (96%), R237 (87%), R245 (72%), R314 (98%), R318 (97%) R320 (48%), K404 (48%), R405 (94%), R413 (72%)

## Methods

**Chemicals**. All chemicals were purchased from Sigma-Aldrich unless specified otherwise.

**Constructs**. Codon-optimized DNA for expression in mammalian cells was obtained from Geneart. The construct used for structure determination, deemed STEAP4$_{EM}$, is truncated by 17 residues at the N-terminus and three residues at the C-terminus. The N-terminal truncation was based on the crystal structure of the OxRD of rat STEAP4. Full-length STEAP4 and STEAP4$_{EM}$ were cloned in a pUPE expression vector (U-Protein Express BV) with a C-terminal GFP-Strep3 tag and a TEV protease site for tag removal. Mutagenesis was performed using the Q5 Site-Directed Mutagenesis Kit (NEB). The primers used in this study are listed in Supplementary Table 1.

**Protein expression and purification**. STEAP4$_{EM}$ was expressed in HEK293 GNTI⁻ suspension cells (provided by U-Protein Express BV) using transient transfection by polyethyleneimine. Cells were typically grown at 37 °C for ~96 h. All subsequent steps were performed at 4 °C, unless stated otherwise. After harvesting, cells were washed in PBS buffer and directly solubilized in lysis buffer containing 50 mM HEPES pH 7.5, 150 mM NaCl, 1% (w/v) n-Dodecyl-β-D-Maltoside (DDM, Anatrace), 0.2% (w/v) Cholesteryl hemi-succinate (CHS), 5 μM hemin, protease inhibitor cocktail (Roche) for 2–3 h. The sample was then subjected to ultracentrifugation at 100,000 × g for 45 min to remove insoluble

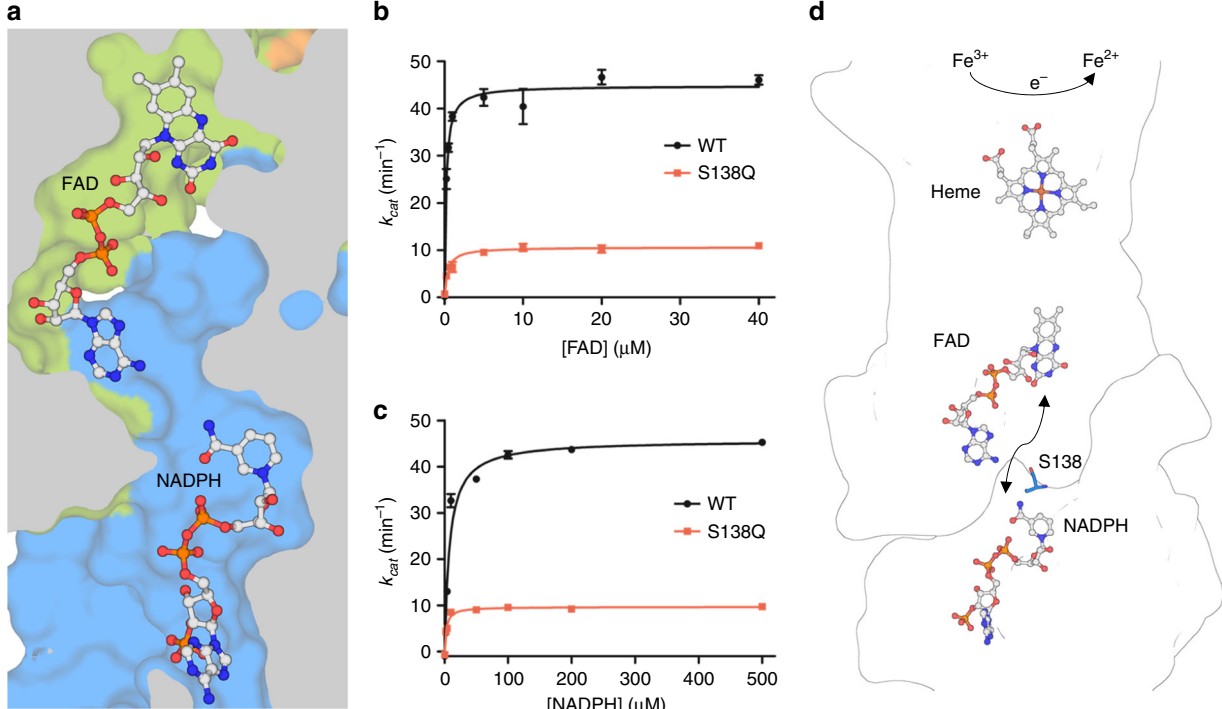

**Fig. 5** Electron-transport pathway in STEAP proteins. **a** Cavity between the FAD and NADPH-binding sites in adjacent STEAP4 protomers. FAD mobility would involve the adenine base flipping out towards the direction of view after which flavin ring diffusion towards NADPH can occur through the cavity. **b** Ferric-reductase activity of detergent-purified, GFP-tagged full-length STEAP4 wildtype (WT) and S138Q mutant with varying FAD concentrations. Error bars represent the standard deviation between triplicate measurements. **c** Ferric-reductase activity of detergent-purified, GFP-tagged full-length STEAP4 WT and S138Q mutant with varying NADPH concentrations. Error bars represent the standard deviation between triplicate measurements. **d** Cofactor arrangement in the intracellular and transmembrane domains of adjacent STEAP4 protomers. Strictly conserved residue S138 is shown in stick representation

membranes and cell debris. The supernatant was filtered and incubated with Streptactin resin (GE Healthcare) for 2 h and the resin was washed with 20 column values of buffer A (50 mM Tris pH 7.8, 150 mM NaCl, 0.05% DDM, 0.01% CHS). Protein was subsequently eluted with buffer A supplemented with 3.5 mM des-thiobiotin. STEAP4-containing fractions were concentrated by an Amicon 100 kDa MW-cutoff concentrator device and incubated with homemade his-tagged TEV-protease (1:4, w/w) for 2–3 h at room temperature. After removal of TEV protease by NiNTA resin (GE Healthcare), the cleaved material was concentrated to a volume of ~450 μl and injected on a Superdex200 increase column (GE Healthcare) pre-equilibrated in 25 mM MES pH 5.5, 200 mM NaCl, 0.08% (w/v) digitonin. Monodisperse trimeric fractions were concentrated to a final concentration of 4.0 mg/ml. Sample purity was assessed with SDS PAGE gel analysis and analytical size-exclusion chromatography.

**Grid preparation**. Concentrated STEAP4$_{EM}$ (4.0 mg/ml) was incubated for several hours before freezing with 0.8 mM NADPH and 0.8 mM FAD for the cofactor-bound state and with 0.8 mM NADP$^+$, 0.8 mM FAD and 0.8 mM Fe$^{3+}$-NTA for the cofactor/substrate-bound state. 2.8 μl of sample was pipetted onto glow-discharged R1.2/1.3 200 mesh Au holey carbon grids (Quantifoil) and then plunge-frozen in liquid ethane with a Vitrobot Mark IV (FEI), blotting for 4 s at 20 °C.

**EM data collection**. For the cofactor-bound dataset, micrographs were collected on a 200 kV Talos Arctica microscope (FEI) equipped with a K2 summit detector (Gatan) and a post column 20 eV energy filter. EPU software (FEI) was employed for automated data collection in superresolution/counting mode. Movies were collected for 6 s in 24 frames with a dose of 1.895 e$^-$/Å$^2$/frame (measured in an empty hole without ice), corresponding to a total dose of 45.5 e$^-$/Å$^2$. The defocus values ranged from 0.8 to 3.0 μm.

For the cofactor/substrate-bound dataset, micrographs were collected in counting mode using Serial EM on a 300 kV Titan Krios microscope (FEI) equipped with a K2 summit detector (Gatan) and a post column 20 eV energy filter. Movies were collected for 12 s in 40 frames with a dose of 1.185 e$^-$/Å$^2$/frame (measured in an empty hole without ice), corresponding to a total dose of 47.4 e$^-$/Å$^2$. The defocus values ranged from 0.8 to 2.0 μm.

**Image processing for the cofactor-bound dataset**. Collected movie stacks were manually inspected and then imported in Relion[30] version 2.1b1. Drift and gain

correction was performed with MotionCor2[31] and GCTF[32] was used to estimate the contrast transfer function for each movie. Movies with a GCTF-estimated resolution of 5.2 Å or worse were discarded. Three tousand four hundred seventy particles were picked manually and 2D classified. The generated classes were used as templates for autopicking[33] in Relion, resulting in 460,932 particles. These particles were subjected to a round of 2D classification after which 39,494 particles were discarded. The remaining particles were imported in Cryosparc[34] and ab initio 3D classified. The class with most features was imported back in Relion and used as initial model (low pass-filtered to 40 Å) for 3D classification into three classes without symmetry applied. The best class (209,075 particles) was selected and refined using 3D auto-refine (with C3 symmetry) which yielded a map with resolution of 4.4 Å. Further sub-classification attempts did not lead to improvements in map quality or resolution. The particles were then subjected to a polishing procedure with all movie frames included. The individual defocus values of these shiny particles were determined by the local CTF option in GCTF, following another 3D Auto-Refinement. Using a post-processing step in Relion, the density map was masked, corrected with the modulation transfer function of the detector and sharpened by applying a negative B-factor of −166 Å$^2$, resulting in a final map at 3.8-Å resolution based on the gold-standard FSC = 0.143 criterion[35]. Local resolution estimations were performed using Relion.

**Image processing for the cofactor/substrate-bound dataset**. Collected movie stacks were manually inspected and then imported in Relion version 2.1. Motion correction and CTF estimation was performed as described for the cofactor-bound dataset. Four thousand four hundred ninety eight particles were picked manually and 2D classified. The generated classes were used as templates for autopicking in Relion, resulting in 1,089,540 particles, which were binned 3×. Using the cofactor-bound map as initial model (low pass-filtered to 40 Å), the particles were 3D classified (C1 symmetry) in five classes (Supplementary Fig. 3c). To obtain the highest-quality particles, the 481,436 particles from the best-looking class were unbinned, and subjected to two additional rounds of 3D classification (C1 symmetry) without image alignment (Supplementary Fig. 3c), through which 226,292 particles were discarded. The remaining 255,144 particles were 3D auto-refined with C3 symmetry applied. Using a post-processing step in Relion, the density map was masked, corrected with the modulation transfer function of the detector and sharpened by applying a negative B-factor of −80 Å$^2$, resulting in a final map at 3.1-Å resolution. Local resolution estimations were performed using Relion.

Focused classifications of the substrate-binding site after particle subtraction[36] in Relion did not result in improved density maps.

**Difference map generation**. To create a difference map between the cofactor/substrate and cofactor-bound reconstructions, both unsharpened, unmasked maps were interpolated onto the grid of the cofactor/substrate-bound map with UCSF Chimera. To allow for direct comparison of both maps, the rotational average of the power spectrum of the higher resolution cofactor/substrate-bound map was then adjusted to be the same as that of the cofactor-bound map using Relion image handler, which assured that the difference map of the two maps is not dominated by different resolutions and imaging parameters of both maps.The cofactor-bound map was then subtracted (using Relion) from the cofactor/substrate-bound map, which yielded a difference map.

**Model fitting and building and refinement**. For the intracellular OxRD (residues 19–195), a model was generated by the SwissProt server based on the crystal structure of the intracellular domain of rat STEAP4 (81% identical, pdb: 2yjz). The generated model was docked in the cofactor-bound map using UCSF Chimera[37]. The TMD was build de novo in this 3.8-Å cofactor bound map with COOT[38]. After obtaining the 3.1-Å cofactor/substrate-bound map, the model was manually adjusted if necessary. The models were iteratively refined manually in Coot and using Phenix[39] real-space refine with non-crystallographic symmetry (NCS) restraints and geometric restraints. Phenix was also used to determine the real-space correlation for each amino acid and to determine the map vs. model Fourier shell correlation, with an atom mask radius of 3.5 Å. The final models comprise residues 19–454 of human STEAP4. All figures were prepared using Pymol (Schrödinger), UCSF Chimera and Prism 5 (GraphPad, La Jolla).

**Surface potential calculation**. Potentials were calculated by APBS[40,41] based on pKa values determined by PROPKA[42]. pKa values were computed at pH 5.5 to match the EM-sample preparation conditions.

**Thermostability assay**. A thermostability assay was employed as previously described[23,24], with minor adjustments. Aliquots of purified STEAP4 were heated over a range of temperatures (4–75 °C) in a thermocycler for 10 min, cooled down and centrifuged to remove aggregates. The supernatant was subsequently injected on a Superdex200 10/300increase equilibrated in 20 mM Tris pH 8, 150 mM NaCl 0.05% DDM, 0.01% CHS, monitoring peak height of heme absorbance (416 nm). Peak heights were normalized to the peak height of the sample incubated at 4 °C and fitted to a dose response equation using GraphPad Prism 5 for determining the melting temperature ($T_m$). Protein was initially purified in 20 mM Tris pH 8.0, 150 mM NaCl, 0.08% digitonin. To find buffer conditions that would lead to a higher melting temperature, we performed the assay at a temperature slightly higher than the $T_m$ (48 °C), while systematically adding either buffer with different pH (spanning 4.0–9.5), salt or cofactor. Several additives strongly improved heme retention at 48 °C, indicating they had a stabilizing effect on STEAP4. The combined addition of 0.5 mM NADPH, 0.5 mM FAD, and 50 mM MES pH 5.5 led to a $T_m$ of 56.3 °C, an increase of 11.0 °C.

**Multi-angle laser light scattering**. Purified, non-TEV cleaved STEAP4$_{EM}$ was injected on a Superdex200 10/300increase column equilibrated in 20 mM Tris pH 8.0, 150 mM NaCl and 0.08% digitonin at a flowrate of 0.75 ml/min. For molecular weight measurements, the column was connected to an online light scattering detector (miniDAWN TREOS, Wyatt Technology) device and a differential refractive index monitor (Shimadzu RID- 10A) on a HPLC system (Shimadzu). Chromatograms were collected and analyzed using the ASTRA software suite (Wyatt). The protein conjugate module was employed for separating the molecular weights of protein and micelle. For STEAP4, a dn/dc of 0.184 ml/g was used based on one predicted N-linked glycan per subunit. For digitonin, we used a previously determined dn/dc value of 0.153 ml/g[43]. Bovine serum albumin was used as a standard.

**Cellular ferric-reductase experiments**. Cells expressing STEAP4 variants were washed in PBS and then resuspended in iron uptake buffer (25 mM MES, 25 mM MOPS pH 7.0, 140 mM NaCl, 5.4 mM KCl, 1.8 mM CaCl$_2$, and 0.8 mM MgCl$_2$) supplemented with 200 μM Fe$^{3+}$-NTA and 400 μM ferrozine. Cells were incubated for 30 min in the dark at 37 °C after which Fe$^{2+}$-ferrozine formation was monitored using a Model 680 microplate reader (Biorad) at 550 nm. The formed Fe$^{2+}$ was quantified using a standard curve[44]. Experiments were performed in triplicate from cells originating from three separate transfections.

**Ferric reductase experiments with purified protein**. STEAP4 purified in SEC buffer (25 mM MES pH 5.5, 200 mM NaCl, 0.08% digitonin) was probed for enzymatic activity using the ferrozine method. Assays using the full-length protein were performed with GFP-tagged STEAP4, whereas the GFP-cleaved variant was used for assays withs STEAP4$_{EM}$. Enzyme (125 nM final concentration) was mixed with 200 μM Fe$^{3+}$-NTA, 10 μM FAD, 800 μM ferrozine and pre-incubated for several minutes at 37 °C. NADPH (75 μM final concentration) was added to start

the reaction. After 1–2 min, Fe$^{2+}$-ferrozine formation was monitored as described above. Michaelis-Menten curves and binding curves were determined by varying the concentration of either Fe$^{3+}$-NTA, FAD or NADPH, respectively, while keeping the other two variables constant. Michaelis-Menten curves and binding curves were fitted using non-linear regression in Graphpad Prism 5.0. To correct for non STEAP4-specific reduction of iron, the absorbance of the same condition without STEAP4 was subtracted for each data point. All reported $K_M$, $K_d$, and $k_{cat}$ values were determined from triplicate experiments. Experiments in which variables were compared were performed on the same day with the same stock solutions. Triplicates were achieved by diluting the same enzyme stock three separate times.

## Data availability

Data supporting the findings of this manuscript are available from the corresponding author upon reasonable request. The relevant cryo-EM density maps of cofactor/substrate-bound and cofactor-bound STEAP4 have been deposited under accession numbers EMDB-0199 and EMDB-0200, respectively. These depositions include unfiltered-half maps, non-sharpened unmasked maps, sharpened masked maps and the difference map generated between the two reconstructions. Model coordinates for both structures have been deposited in the Protein Data Bank under accession numbers 6HCY and 6HD1.

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

## Acknowledgements

We acknowledge Crystal and Structural Chemistry lab members for many helpful discussions. We kindly thank W. Hemrika (Utrecht Protein Express BV) for HEK-cell cultures and the Utrecht EM-square facility for assistance at the Talos Arctica microscope in Utrecht. The cofactor/substrate-bound dataset was acquired at the cryo-electron microscopy facility of the European Molecular Biology Laboratory (EMBL) in Heidelberg. This work has been supported by the Netherlands Organization for Scientific Research (NWO), Fund NCI Technology Area (project no. 731.015.201), and iNEXT (project number 653706) funded by the Horizon 2020 programme of the European Union. Research on NOXs in the laboratory of A.M. is supported by the Associazione Italiana per la Ricerca sul Cancro (AIRC; IG19808).

## Author contributions

W.O. and P.G. designed the project. W.O., R.N.P.R. and J.G. screened initial constructs. W.O. carried out protein purification and biochemical assays. W.O. and L.S.vB. did the EM sample preparation, data collection, and data processing. W.O. performed the model building and model refinement. W.O., F.F., A.M. and P.G. analyzed the data. J.G. did the bulk of the molecular cloning. W.O. and P.G. wrote the manuscript, with critical input from all authors.

## Additional information

**Competing interests:** The authors declare no competing interests.

