## [Peer Review File · Nature Communications]

Reviewers' comments:

Reviewer #1 (Remarks to the Author):

Reported in PNAS in 1999, the Afar group first identified the gene family christened as STEAP – six-transmembrane epithelial antigen of the prostate – a gene up-regulated in a number of cancers. In short, the connection of the STEAP proteins and metastatic disease has been a topic of investigation for nearly a score of years. Indeed, one can read reviews to that effect as early as 2012 (Mol Can Res 10:573, Biol Cell 104:641) – “STEAP proteins as oncogenic targets.” The fact that the STEAP proteins are metalloreductases was established by Mark Fleming in 2006; this work was done since STEAP proteins resemble the well-characterized reductases from fungi, for example, that are involved in copper and iron uptake in those organisms, and the Dcytb protein then known to support iron uptake from the intestinal lumen. Fleming subsequently found that STEAP3 was essential to the mobilization of transferrin iron in the endosome and together with Martin Lawrence reported the structure of the membrane proximal oxidoreductase domain of STEAP3 (J Biol Chem 288:20668). In 2015, Lawrence followed up with a detailed examination of the mechanism of the shuttling of electrons from this domain, through the membrane and to the higher valent metal ion substrate (J. Biol Chem 290:22558). Similar data for STEAP 1 has been published, as well, demonstrating conservation of mechanism among the STEAP proteins (Biochemistry 55:6673). For the authors of this manuscript to state in the third sentence of their abstract that the “mechanism of these proteins is unclear” is simply wrong and misleading, implying to the reader that the manuscript to follow is more innovative and significant than it can honestly claim to be. This is unfortunate since the work has much to offer that is new.

What this work has to offer is a construction of the complete protein, transmembrane domain along with the oxidatively catalytic cytoplasmic domain as well as a model for how the metal substrate is bound at the exocytosolic face of the reductase. These models are based on a pair of cryo-EM data sets that provide an ~ 3 Å map, part of which was ‘beta-tested’ against the crystallographically-determined structures of the cytoplasmic domains (noted above). I am far from an expert on cryo-EM methodology, but the experimental approach (choosing STEAP target protein, protein expression and characterization, and sample preparation) was above reproach. In short, experimentally this was excellent work presented in a concise, compelling manner.

A research article can provide confirmatory and/or novel information. From the point of view of connecting function to structure, this paper provides a bit of both. With respect to the binding of heme, an essential prosthetic group, previous mutagenesis indicated this occurred between His residues located on TM helices 3 and 5; this work confirms this. FAD binding was similarly confirmed. What was new in this work was data that appear to locate the ferric iron substrate binding site positioned adjacent to the heme edge oriented towards the exocytosolic ends of the TM domain bundle. The data also indicated that bound at this location was the ferric iron complex and not the ‘free’ ion. Although not discussed, this result was consistent with kinetic studies that indicated the ferric iron complex was the substrate based on consideration of redox potential and coordination complex stability. In this way, the one thing this work does is to put the wealth of kinetic and biophysical data on STEAP proteins into a fairly complete and rigorous structural context and on that basis this work does offer a novel contribution to the field. The same can be said for the further elucidation of the binding of the protein’s FAD co-factor; previous work indicated its binding to the structurally-characterized cytoplasmic domain, while the work here demonstrates a contribution from cytoplasmic-facing TM domains, as well. Again, what this paper does is provide a picture of function in the context of structure that is significantly fuller than that afforded by previous studies limited to models of the protein’s TM domain.

I can recommend this paper for consideration with respect to publication in Nature Communications. The reader would benefit, however, if the authors would do a more thoughtful and thorough job of directly connecting their model for electron transfers to the excellent, robust and more kinetically rigorous data one finds in the literature. By doing so, they will be truly

making a contribution to the field.

Reviewer #2 (Remarks to the Author):

Members of the STEAP (Six transmembrane epithelial antigen of the prostate) protein family are ferric and cupric reductases that reduce extracellular Fe³⁺ and Cu²⁺ as a prerequisite for transport of these metals into the cell. Because the activity of Steap proteins provide iron needed for rapid cell growth, Steap proteins are routinely up regulated in many cancers. And like other proteins involved in iron transport, they are also implicated in oxidative stress and innate immunity. Steap4, in particular, is commonly upregulated in a large number of inflammatory disease states where, paradoxically, it is thought to play a potentially important role in the cellular response to inflammatory stress. For these reasons, there is widespread interest in the molecular mechanism by which electrons from intracellular NADPH are used to reduce extracellular iron and copper.

This manuscript presents the structure of Steap4, clearly shows that it is a domain swapped trimer, reports the structure of the transmembrane domain and its relationship to the NADPH oxidase domain whose structure had been previously determined, and provides a detailed molecular framework for understanding electron flow from intracellular NADPH, through Steap4 bound FAD and heme, to the extracellular metal. This structure of human Steap4 is thus a critical advance that will be widely appreciated by a members of various fields, including those working in cancer biology, inflammation, iron and copper homeostasis, etc. In addition, the work is technically well done and the paper is well written; it is concise and clear in all aspects. Relative to criteria for publication in Nature Communications, it is clear that this manuscript reports novel results of extreme importance with strong conclusions to scientists in several different specific fields, and will also be of interest to researchers in a number of other related disciplines. I congratulate the authors on their fine work.

I have only one minor suggestion. On page 6 of the manuscript, I believe it is appropriate for the authors to acknowledge that reference 14 also concluded the cytosolic heme-binding site of the FRD superfamily evolved into the Steap protein flavin binding site.

Martin Lawrence
Montana State University

Reviewer #3 (Remarks to the Author):

In this manuscript, Oosterheert et al. describe the single-particle cryo-EM structure of biologically active STEAP4, which reduces Fe³⁺ and Cu²⁺ ions for subsequent transmembrane transport by specific metal transporters. After optimizing expression, purification and specimen preparation, the authors determined cryo-EM maps of STEAP4 with its co-factors, NADPH and FAD, and with co-factors and substrate, NADP⁺, FAD and Fe³⁺-NTA, at resolutions of 3.8 and 3.1 Å, respectively. Based on their structures, the authors propose that STEAPs facilitate transmembrane electron transport using a single heme, and comparison with NOX5, which contains two hemes, reveals that one of the heme-binding sites evolved to bind FAD in STEAPs. In their models, the nicotinamide ring of NADPH does not stack to a flavin ring, which is normally the case for electron transfer, which leads the authors to propose that the FAD may be mobile. Mutagenesis of a highly conserved residue, S138, between the two co-factor-binding sites to glutamine greatly reduces the ferric reductase activity of STEAP5, consistent with the larger glutamine side chain interfering with the diffusion of FAD.

This is a very nice and high-quality study that provides structural and mechanistic insights into the reduction of Fe³⁺ and Cu²⁺ ions by STEAPs. While the presented structures are strong, the

obvious weakness of the study lies in the remaining uncertainties, but these should not prevent publication of this work. The three areas that would benefit from additional information and clarification are:

1. The proposed model for the transmembrane electron transport by STEAPs appears to depend critically on residue F359, which connects the heme to the FAD. What is the conservation of this residue and did the authors attempt to mutate this residue to interrupt electron transport? How does the F359-mediated electron transfer compare to how electrons are transferred between the two hemes in NOX5?
2. The authors suggest that the weaker-than-expected density for the substrate, Fe³⁺-NTA, may be due to heterogeneous coordination or perturbation by the electron beam. Another, maybe more commonly used, explanation for weak density would be low occupancy. While the authors presumably have a good reason to exclude this possibility, this reason should be stated. It would also be good to show the density for the substrate in the regular map as an Extended Data figure. Since the protein is so rigid, did the authors attempt to perform a classification with density subtraction focused on the area that contains the difference density to see whether they can obtain better density features in this region? Also, how conserved are the basic amino acids that the authors suggest form a second coordination shell that may facilitate metal reduction?
3. The idea that the FAD may be mobile is interesting, but a reduction in ferric reductase activity upon mutation of S138 is rather indirect evidence for this idea. What would cause the FAD to move as it seems to be in the same position with and without substrate? Is the density for the FAD weaker than that for other co-factors or neighboring residues, which would support the notion that the FAD is mobile? Also, did the authors attempt a focused classification to see whether they can observe the FAD in different positions? Some additional evidence in support of FAD mobility would greatly strengthen the paper.

Some minor points:

The SDS-PAGE gel in Extended Fig. 1b shows an appreciable number of additional bands and it would be good to know what these are, i.e., a mass spectrometry analysis of the additional bands would identify which ones represent degradation products, oligomers, and contaminants.

The description of the image processing procedure lacks almost all details and needs to be greatly expanded (e.g., how many particles were discarded after 2D classification, 3D classification was done into how many classes, how many particles did the final reconstruction contain etc. etc. etc.). The authors should also show all the classes that resulted from 3D classification. Moreover, what do the authors mean with "Using `reliion_image_handler`, the power spectrum of the higher resolution cofactor/substrate-bound map was adjusted to be the same as for the cofactor-bound map"? Why was the power spectrum adjusted and what was adjusted – resolution or intensity? Why was the power spectrum used and not the Fourier transform? What program was used to subtract the maps from each other? The authors state that they calculated "map to model Fourier shell correlations", but it is not clear whether they show the resulting curves. Instead, the authors show FSC plots of high-resolution phase-randomized, unmasked and masked half maps, but neither the details of this procedure nor the results are described.

If the authors believe that the resolution difference between the two maps is simply due to the microscope that was used for data collection, why do they not just collect a data set of STEAP4 without substrate with the Krios instrument? It is certainly not necessary, since both structures are essentially identical, but it does seem a bit odd, since many ~ 3 Å structures have recently been determined with Arctica instruments. Did maybe the different size of the data sets have an effect on the final resolution? It is impossible to figure out, since the authors not even provide the number of particle images in the final density maps (unless I missed the numbers).

Why do the authors consider helices h1 and h6 not to be part of the transmembrane core? The distinction of helices h1 and h6 from helices h2 to h5 seems a bit random without further explanation.

Extended Data Fig. 5c does not do a good job illustrating the full density of the bound lipid nor does it show particularly well where the lipid is bound to the protein. This panel needs to be improved and possibly split into two panels (overview and detail view).

The central heme iron is not really visible in Fig. 2a and b, so the authors may want to also refer to Fig. 2e for this purpose and label the iron in this panel.

It can be confusing when the authors refer to the extracellular and cytoplasmic heme of NOX5, since both hemes are transmembrane. Better to refer to these hemes as being located in the extracellular and cytoplasmic leaflets of the membrane.

FAD and NADPH should be labeled in Fig. 5a.

Rebuttal to Reviewers' comments:

We thank the reviewers for their positive and constructive critiques.

Changes to document and comments made are marked e.g. "Reply 1.1" (referring the Reviewer #1 item #1, etc.) as described below.

Reviewer #1:

[1.1] Reported in PNAS in 1999, the Afar group first identified the gene family christened as STEAP ... to state in the third sentence of their abstract that the "mechanism of these proteins is unclear" is simply wrong and misleading, implying to the reader that the manuscript to follow is more innovative and significant that it can honestly claim to be. This is unfortunate since the work has much to offer that is new.

We agree with Reviewer 1 that previous studies provided mechanistic insights into electron transfer. It was not our intention to downplay previous literature.

Reply 1.1: We adjusted the sentence in Abstract (lines 3-5):
"However, the structural basis for STEAP-catalyzed electron transfer through an array of cofactors to metals at the membrane luminal side remains elusive."

The reviewer lists a number of previous publications. All these publications were referenced in the previous manuscript. It is unclear to us if the Reviewer criticizes the introduction section of our manuscript or if she/he just summarizes previous literature. To be sure, we explicitly reply to these comments.

[1.1a] Reported in PNAS in 1999, the Afar group first identified the gene family christened as STEAP – six-transmembrane epithelial antigen of the prostate – a gene up-regulated in a number of cancers.

Reply 1.1a: To emphasize the first identification, we added to the Introduction (page 2, paragraph 2, lines 3-4): "First identified as antigens upregulated on the membranes of prostate cancer², ..."

[1.1b] In short, the connection of the STEAP proteins and metastatic disease has been a topic of investigation for nearly a score of years. Indeed, one can read reviews to that effect as early as 2012 (Mol Can Res 10:573, Biol Cell 104:641) – "STEAP proteins as oncogenic targets."

Reply 1.1b: In the manuscript we stated: "STEAPs are associated with metabolic diseases⁶⁻⁸ and are over-expressed in [several] human cancers^{2,9-11}, underlining their physiological function in maintaining cellular iron homeostasis^{6,12} ...", which included a reference (number 11) to Grunewald *et al.* (Biol Cell 104:641).

We also stated: "Despite the association of STEAP proteins with severe diseases and although they are potential therapeutic targets¹⁶⁻¹⁸, structural information...", which referred to (number 18) Gomes *et al.* (Mol Can Res 10:573).

[1.1c] The fact that the STEAP proteins are metalloredoxases was established by Mark Fleming in 2006; this work was done since STEAP proteins resemble the well-characterized reductases from fungi, for example, that are involved in copper and iron uptake in those organisms, and the Dcytb protein then known to support iron uptake from the intestinal lumen.

Reply 1.1c: In the manuscript we stated: “[The] enzymes of the six-transmembrane epithelial antigen of the prostate (STEAP) protein family catalyze iron(III) reduction³⁻⁵ ...”, where we reference the work from Fleming in 2005 and 2006 (ref. 3 and 4).

[1.1d] Fleming subsequently found that STEAP3 was essential to the mobilization of transferrin iron in the endosome and together with Martin Lawrence reported the structure of the membrane proximal oxidoreductase domain of STEAP3 (J Biol Chem 288:20668).

Reply 1.1d:

We referenced the papers (ref 19 and ref 14) that describe the crystal structures of the membrane proximal oxidoreductase domains of human STEAP3 (PNAS 105:7410) and rat STEAP4 (J. Biol. Chem. 288:20668) several times in the main part of the text. However, we now added to the last sentence of paragraph 3 (page 3) in the introduction: “structural information is only available for the cytoplasmic NADPH-binding OxRD of human STEAP3¹⁹ and rat STEAP4¹⁴, but is absent for the cofactor and substrate-binding TMD of any STEAP”.

[1.1e] In 2015, Lawrence followed up with a detailed examination of the mechanism of the shuttling of electrons from this domain, through the membrane and to the higher valent metal ion substrate (J. Biol Chem 290:22558).

Reply 1.1e: We referred to this publication (ref 15) in the introduction: “Previous studies suggest that a FAD cofactor might substitute and mimic the second, missing heme in STEAPs, because FAD interacts with the intracellular loops of the TMD^{14,15}”. We now also added a discussion section in which we compare the work of Lawrence with our structure: “Residues H304 and H397, which coordinate the central heme iron (Fig. 2a, b), were reported by several studies to be imperative for heme binding and metal-ion reduction by STEAP proteins^{9,15}. Strictly conserved residues R290, K291 and Q383 were identified through mutagenesis studies to participate in FAD binding¹⁵ ...”

[1.1f] Similar data for STEAP 1 has been published, as well, demonstrating conservation of mechanism among the STEAP proteins (Biochemistry 55:6673).

Reply 1.1f: We referenced this paper (ref 5) several times. In the introduction we stated: “enzymes of the six-transmembrane epithelial antigen of the prostate (STEAP) protein family catalyze iron(III) reduction³⁻⁵ ..” because ref 5 showed that also STEAP1 is able to function as ferric reductase. In the result section (“biochemical characterization” subheader) we addressed the oligomeric state of STEAPs: “This finding is consistent with MALLS experiments on STEAP1⁵, but ...”.

What this work has to offer is a construction of the complete protein, In short, experimentally this was excellent work presented in a concise, compelling manner.

No comments.

A research article can provide confirmatory and/or novel information. ... The reader would benefit, however, if the authors would do a more thoughtful and thorough job of directly connecting their model for electron transfers to the excellent, robust and more kinetically rigorous data one finds in the literature. By doing so, they will be truly making a contribution to the field.

The reviewer lists a number of previously published findings. We added a Discussion section to the manuscript to explicitly link our observations with these previous findings.

[1.2a] With respect to the binding of heme, an essential prosthetic group, previous mutagenesis indicated this occurred between His residues located on TM helices 3 and 5; this work confirms this. FAD binding was similarly confirmed.

Reply 1.2a: We added the following sentences to the discussion section: “Residues H304 and H397, which coordinate the central heme iron (Fig. 2a, b), were reported by several studies to be imperative for heme binding and metal-ion reduction by STEAP proteins^{9,15}. Furthermore, strictly conserved residues R290, K291 and Q383 were identified through mutagenesis studies to participate in FAD binding¹⁵; the cryo-EM structures confirm a direct interaction between FAD and these residues and, in addition, reveal several other FAD-interacting amino acids (Fig. 2d)”.

[1.2b] ... bound at this location was the ferric iron complex and not the ‘free’ ion. Although not discussed, this result was consistent with kinetic studies that indicated the ferric iron complex was the substrate based on consideration of redox potential and coordination complex stability.

Reply 1.2b: The manuscript indeed did not refer to reference 5 (Biochemistry 55:6679) with regard to a substrate-binding model. We therefore added a sentence to the discussion: “This structural observation is in agreement with findings from an earlier electrochemical study, in which the same conclusion was derived based on the measured redox potential of detergent-purified STEAP1⁵”.

Reviewer #2 (Martin Lawrence, Montana State University):

[2] I have only one minor suggestion. On page 6 of the manuscript, I believe it is appropriate for the authors to acknowledge that reference 14 also concluded the cytosolic heme-binding site of the FRD superfamily evolved into the Steap protein flavin binding site.

Reply 2: We changed the sentence on page 6, which now states: “Our structural analysis now supports the conclusion that the heme-binding site in the cytoplasmic membrane leaflet evolved into a flavin-binding site¹⁴.”

Reviewer #3:

The three areas that would benefit from additional information and clarification are:

[3.1] The proposed model for the transmembrane electron transport by STEAPs appears to depend critically on residue F359, which connects the heme to the FAD. What is the conservation of this residue and did the authors attempt to mutate this residue to interrupt electron transport? How does the F359-mediated electron transfer compare to how electrons are transferred between the two hemes in NOX5?

Reply 3.1: We added a paragraph in the Discussion section regarding residue F359: “Transfer of electrons across the membrane may follow a path through residue F359 positioned in between FAD and heme. STEAP orthologs harbor phenylalanines, tyrosines or leucines at the equivalent position, suggesting that the presence of an aromatic residue is not critical for transporting electrons across the membrane and that alternative routes through the protein may exist. Likewise, the residue located in between the diheme motif in the TMD of NOX5 is not strictly conserved as aromatic residue within the NOX family²⁶”.

[3.2a] The authors suggest that the weaker-than-expected density for the substrate, Fe³⁺-NTA, may be due to heterogeneous coordination or perturbation by the electron beam. Another, maybe more commonly used, explanation for weak density would be low occupancy. While the authors presumably have a good reason to exclude this possibility, this reason should be stated.

Reply 3.2a: The reason for not stating low occupancy as explanation for the weak density, is that Fe³⁺-NTA was added to the sample in concentrations higher than 50x the K_M. We now state this in the manuscript on page 6: “...even though the cryo-EM sample was supplemented with Fe³⁺-NTA at a concentration 50x higher than the K_M (Fig. 1a)”.

[3.2b] It would also be good to show the density for the substrate in the regular map as an Extended Data figure.

Reply 3.2b: We created Supplemental Figure 8 to show the substrate density in the regular, unsharpened map. The density for both substrate and heme is shown in the same figure at two different contour levels (6σ and 8σ) to emphasize that the substrate density is weaker than expected.

[3.2c] Since the protein is so rigid, did the authors attempt to perform a classification with density subtraction focused on the area that contains the difference density to see whether they can obtain better density features in this region?

Reply 3.2c: We did attempt focussed classifications after density subtraction in Relion as described by Bai et al. (eLife 2015;4:e11182), but that did not result in stronger or higher quality density at the substrate-binding site. This is now incorporated in the method section

on page 11: “Focused classifications of the substrate-binding site after particle subtraction³⁶ in Relion did not result in improved electron density maps”.

[3.2d] Also, how conserved are the basic amino acids that the authors suggest form a second coordination shell that may facilitate metal reduction?

Reply 3.2d: The conservation of the basic amino acid ring is stated in the legend of Fig. 4b: “The shown residues display variable conservation as basic amino acids (percentage in brackets) throughout 607 STEAP orthologs¹⁵: R221 (96%), R237 (87%), R245 (72%), R314 (98%), R318 (97%) R320 (48%), K404 (48%), R405 (94%), R413 (72%)”. We now added a section in the discussion in which we refer to this: “Sequence comparison of 607 STEAP orthologs (see Fig. 4b legend) indicates that not all nine basic residues in the STEAP4 substrate-binding cavity in STEAP4 are highly conserved. Nevertheless, all human STEAPs harbor at least 6 positively-charged residues at the substrate-binding site, suggesting a common mechanism for substrate binding and reduction”.

[3.3] The idea that the FAD may be mobile is interesting, but a reduction in ferric reductase activity upon mutation of S138 is rather indirect evidence for this idea. What would cause the FAD to move as it seems to be in the same position with and without substrate? Is the density for the FAD weaker than that for other co-factors or neighboring residues, which would support the notion that the FAD is mobile? Also, did the authors attempt a focused classification to see whether they can observe the FAD in different positions? Some additional evidence in support of FAD mobility would greatly strengthen the paper.

Reply 3.3: We fully agree with the reviewer that the results presented in our manuscript do not show direct evidence for the diffusion of FAD. The density for FAD is strong and focussed classifications do not provide proof for an alternative conformation of the cofactor. However, besides the mutation of S138, the manuscript describes other indications that support the hypothesis of a mobile FAD in the section ‘Electron transfer from NADPH to FAD’ on page 7, namely:

(1) the crystal structure of the similar OxRD of FNO with F₄₂₀ stacked to NADPH:

“Superimposition of the STEAP4 OxRD onto FNO (rmsd = 0.9 Å for 120 Cα atoms), which has NADPH bound in a similar orientation with a stacked flavin-like F₄₂₀ molecule, suggests a potential NADPH-stacking site for a diffusible FAD in STEAP4 (Supplementary Fig. 11)”.

(2) The (non-physiological relevant) activity with FMN:

“We performed enzymatic assays using flavin-mononucleotide (FMN) instead of FAD, which revealed that FMN may act as a cofactor in STEAP4-catalyzed iron reduction, albeit only at high, non-physiological concentrations (Supplementary Fig. 10)”

(3) the moderate K_d of STEAP4 for FAD:

“Taken together with the moderate K_d of FAD, this suggests that FAD could be mobile”.

We envision that obtaining direct evidence for FAD diffusion through either structural or spectroscopic techniques will require multiple non-trivial experiments. Testing our FAD-diffusion hypothesis was therefore just not within the scope of the research described in the current manuscript, although it will remain an interesting subject for future research.

Some minor points:

[3.4] The SDS-PAGE gel in Extended Fig. 1b shows an appreciable number of additional bands and it would be good to know what these are, i.e., a mass spectrometry analysis of the additional bands would identify which ones represent degradation products, oligomers, and contaminants.

Reply 3.4: We now labelled the major STEAP4-band for clarity in Supplementary Figure 1b. The gel indeed shows some additional bands. However, we do not think that a mass-spectrometry analysis would provide crucial new insights for the current study, since we did not encounter problems with degradation products and/or contaminants in the EM image processing.

[3.5a] The description of the image processing procedure lacks almost all details and needs to be greatly expanded (e.g., how many particles were discarded after 2D classification, 3D classification was done into how many classes, how many particles did the final reconstruction contain etc. etc. etc.). The authors should also show all the classes that resulted from 3D classification.

Reply 3.5a: We expanded the description of image processing steps. The image processing described in the methods section is now split in three separate panels (1) processing of cofactor-bound dataset, (2) processing of cofactor/substrate-bound dataset, (3) generation of difference map. 3D classification figures have been added to Supplementary Figures 2c and 3c, with particle numbers provided.

[3.5b] Moreover, what do the authors mean with “Using relion_image_handler, the power spectrum of the higher resolution cofactor/substrate-bound map was adjusted to be the same as for the cofactor-bound map”? Why was the power spectrum adjusted and what was adjusted – resolution or intensity? Why was the power spectrum used and not the Fourier transform? What program was used to subtract the maps from each other?

Reply 3.5b: The power spectrum of the cofactor/substrate-bound map was adjusted to be the same as for the cofactor-bound map, to allow for a direct comparison between both maps. The idea of the normalization implemented in Relion is that systematic modulations of the maps in Fourier space, which can only partially be approximately compensated for in the reconstruction process. The phases are not affected by the different microscopes and detectors and hence not adjusted. Factors affecting the modulation of amplitudes include different magnifications (1.03 vs. 0.81 Å pixel size) and modulation transfer functions of the detector (K2 at 200 vs. 300 kV) as well as properties of the electron source (e.g., 200 vs. 300 kV voltage). Thus, the procedure adjusts the intensities of a target map to a reference. Specifically, the Fourier transforms of target and reference are divided into shells, as also done for FSC calculation, and the target shells are scaled by the ratio of the mean values of the amplitudes in reference and target. As a by-product, the resolution is also adjusted because shells that have zero intensity in the cofactor-bound reference map will also be set to zero in the higher-resolution cofactor/substrate-bound target map. Using this procedure, the amplitudes of the cofactor/substrate-bound were scaled in each shell to the same average value as the cofactor-bound map.

The method section on this is updated: "To create a difference map between the cofactor/substrate and cofactor-bound reconstructions, both unsharpened, unmasked maps were interpolated onto the grid of the cofactor/substrate-bound map with UCSF Chimera. To allow for direct comparison of both maps, the rotational average of the power spectrum of the higher resolution cofactor/substrate-bound map was then adjusted to be the same as that of the cofactor-bound map using Relion image handler, which assured that the difference map of the two maps is not dominated by different resolutions and imaging parameters of both maps."

Relion was used to subtract the maps from each other. We updated the method section: "The cofactor-bound map was then subtracted (using Relion) from the cofactor/substrate-bound map..."

[3.5c] The authors state that they calculated "map to model Fourier shell correlations", but it is not clear whether they show the resulting curves. Instead, the authors show FSC plots of high-resolution phase-randomized, unmasked and masked half maps, but neither the details of this procedure nor the results are described.

Reply 3.5c: The map vs. model Fourier shell correlations are shown in Supplemental Fig. 4c and Supplemental Fig. 5e. The FSC plots of high-resolution phase-randomized, unmasked and masked half maps in Supplemental Fig. 2g and Supplemental Fig. 3g depict the standard output in Relion; The phase-randomization procedure is described by Chen et al. (Ultramicroscopy 2013; 135:24-35). Phases are randomized beyond the frequency where the FSC of the unmasked half-maps = 0.8. This procedure is used to fine-tune the effect of the mask on the FSC and to prevent overfitting. The phase-randomized FSC curves for our two datasets drop sharply, indicating that no large correction for effects of the mask is required. We added the following sentence to figure legend of Supplemental Fig. 2g: "The phase-randomized FSC curve was generated by Relion through a described procedure⁴⁵" – citing the Ultramicroscopy paper.

[3.6] If the authors believe that the resolution difference between the two maps is simply due to the microscope that was used for data collection, why do they not just collect a data set of STEAP4 without substrate with the Krios instrument? It is certainly not necessary, since both structures are essentially identical, but it does seem a bit odd, since many ~3 Å structures have recently been determined with Arctica instruments. Did maybe the different size of the data sets have an effect on the final resolution? It is impossible to figure out, since the authors not even provide the number of particle images in the final density maps (unless I missed the numbers).

Reply 3.6: The initial and final particle numbers are stated in Table 1, but now have been added to the methods section and in Extended Figures 2 and 3. Because the final reconstructions contain more than 200k particles for the cofactor-bound dataset and the cofactor/substrate-bound dataset, we do not think that the amount of particles is a limiting factor.

Utrecht University does not own a Titan Krios microscope; hence we have limited access to 300 kV microscopes at external institutions. The obtained resolution of 3.8 Å on a 200 kV

microscope for a small membrane protein with an ordered mass of ~150 kDa is comparable to several depositions in the pdb (e.g. <https://www.nature.com/articles/s41594-018-0076-y>).

[3.7] Why do the authors consider helices h1 and h6 not to be part of the transmembrane core? The distinction of helices h1 and h6 from helices h2 to h5 seems a bit random without further explanation.

Reply 3.7: We agree that the term ‘transmembrane core’ on page 4 is confusing and changed it to ‘cofactor-binding core’ in the manuscript: “... with helices h2, h3, h4 and h5 forming the cofactor-binding core of the protein that spans the (~32-Å) membrane bilayer (Fig. 1d)”.

[3.8] Extended Data Fig. 5c does not do a good job illustrating the full density of the bound lipid nor does it show particularly well where the lipid is bound to the protein. This panel needs to be improved and possibly split into two panels (overview and detail view).

Reply 3.8: We split Extended Fig. 5c into two separate panels, depicting both the location of the lipids in the trimeric structure as well as the amino acids that orient towards the lipid headgroup.

[3.9] The central heme iron is not really visible in Fig. 2a and b, so the authors may want to also refer to Fig. 2e for this purpose and label the iron in this panel.

Reply 3.9: The text on page 5 has been changed to also refer to Fig. 2e, in which the central heme iron is now labelled: “Strictly conserved residues H304 and H397 of helices h3 and h5, respectively, coordinate the central heme iron, defining a hexa-coordinated porphyrin (Fig. 2a, b, e)”.

[3.10] It can be confusing when the authors refer to the extracellular and cytoplasmic heme of NOX5, since both hemes are transmembrane. Better to refer to these hemes as being located in the extracellular and cytoplasmic leaflets of the membrane.

Reply 3.10: ‘Extracellular/cytoplasmic heme’ has been changed to ‘heme in extracellular/cytoplasmic membrane leaflet’ on page 6: “...the heme of STEAP4 has a comparable orientation and coordination as the heme present in the extracellular membrane leaflet of NOX5 (Fig. 3b)”.

[3.11] FAD and NADPH should be labeled in Fig. 5a.

Reply 3.11: We now added labels to Fig. 5a.

REVIEWERS' COMMENTS:

Reviewer #1 (Remarks to the Author):

The authors have reviewed my comments and addressed my concerns.

Reviewer #3 (Remarks to the Author):

The authors have addressed all my concerns and queries and the manuscript is now suitable for publication. One minor note, the authors occasionally refer to "electron density map" rather than just "density map", which should be changed.

REVIEWERS' COMMENTS:

Reviewer #1 (Remarks to the Author):

The authors have reviewed my comments and addressed my concerns.

No further comments.

Reviewer #3 (Remarks to the Author):

The authors have addressed all my concerns and queries and the manuscript is now suitable for publication. One minor note, the authors occasionally refer to "electron density map" rather than just "density map", which should be changed.

We removed the term "electron density map" from the manuscript and replaced it with "density map".